# Cloud-Edge Collaborative Defect Detection Based on Efficient Yolo Networks and Incremental Learning

**DOI:** 10.3390/s24185921

**Published:** 2024-09-12

**Authors:** Zhenwu Lei, Yue Zhang, Jing Wang, Meng Zhou

**Affiliations:** The School of Electrical and Control Engineering, North China University of Technology, Beijing 100144, China; leizhenwu@ncut.edu.cn (Z.L.); zy18811743082@163.com (Y.Z.); zhoumeng@ncut.edu.cn (M.Z.)

**Keywords:** cloud-edge collaboration, lightweight YoloV5, incremental learning, defect detection, electronics manufacturing

## Abstract

Defect detection constitutes one of the most crucial processes in industrial production. With a continuous increase in the number of defect categories and samples, the defect detection model underpinned by deep learning finds it challenging to expand to new categories, and the accuracy and real-time performance of product defect detection are also confronted with severe challenges. This paper addresses the problem of insufficient detection accuracy of existing lightweight models on resource-constrained edge devices by presenting a new lightweight YoloV5 model, which integrates four modules, SCDown, GhostConv, RepNCSPELAN4, and ScalSeq. Here, this paper abbreviates it as SGRS-YoloV5n. Through the incorporation of these modules, the model notably enhances feature extraction and computational efficiency while reducing the model size and computational load, making it more conducive for deployment on edge devices. Furthermore, a cloud-edge collaborative defect detection system is constructed to improve detection accuracy and efficiency through initial detection by edge devices, followed by additional inspection by cloud servers. An incremental learning mechanism is also introduced, enabling the model to adapt promptly to new defect categories and update its parameters accordingly. Experimental results reveal that the SGRS-YoloV5n model exhibits superior detection accuracy and real-time performance, validating its value and stability for deployment in resource-constrained environments. This system presents a novel solution for achieving efficient and accurate real-time defect detection.

## 1. Introduction

With theadvancement of Industry 4.0, the manufacturing industry has increased its requirements for product quality inspection as it moves towards greater intelligence and automation. In electronics manufacturing, defect detection is crucial for product reliability and safety [1]. Traditional manual inspections are inefficient and prone to errors, failing to meet modern manufacturing needs for accuracy [2].

Recently, automated detection methods using computer vision, especially deep learning, have shown significant results in complex environments [3]. The YOLO series models are widely used in target detection for their real-time performance and high accuracy. However, deploying YOLO models on resource-constrained edge devices presents challenges in computational power and storage. Lightweight models, such as MobileNet [4] and ShuffleNet [5], although optimized in terms of computation and model size, are often inferior to full-size models in terms of detection accuracy. Achieving lightweight and efficient models while ensuring high detection accuracy remains a significant research challenge. Relying solely on edge devices may not handle all complex or rare defects, affecting overall inspection results, whereas relying entirely on cloud computing may encounter bottlenecks in network latency and data transmission [6].

Research addressing this area has focused on improving detection accuracy and efficiency. Recently, many novel target detection models and techniques have been introduced to address these challenges. For example, YOLOv7 and YOLOv8 have been optimized in terms of detection speed and accuracy, and have improved circuit board defect detection by introducing more efficient network architectures and data enhancement techniques [7]. In addition to the YOLO series of models, other deep learning models have also achieved significant results in circuit board defect detection. EfficientDet-D7x [8] significantly reduces the computational effort while ensuring the detection accuracy by means of an efficient feature pyramid network and composite scaling techniques. Deformable DETR-v2 [9] combines deformable convolutional and Transformer architecture, which enhances the model’s adaptability to complex shapes and locations and improves the detection performance. Libra R-CNN [10] improves defect detection by balancing the features of each layer of the feature pyramid. These methods improve the accuracy and efficiency of circuit board component defect detection to different degrees. MobileNetV4 [11] further reduces the amount of computation and the number of parameters by introducing new activation functions and module structures while maintaining high detection accuracy. ShuffleNetV2 [12] improves computational efficiency by optimizing channel segmentation and information flow. Model compression and acceleration techniques are also widely used on resource-constrained devices. Pruning, quantization, and knowledge distillation are commonly used model compression techniques. The pruning technique reduces computation and storage requirements by removing redundant network connections [13]; the quantization technique reduces computation and storage costs by lowering the accuracy of model parameters (e.g., 8-bit quantization) [14]; and the knowledge distillation technique enables small models to be more computationally efficient while maintaining a high level of accuracy by transferring the knowledge of a large model to a small model [15]. In addition, efficient convolutional modules also play an important role in optimizing model performance. For example, GhostNetv2 [16] improves computational efficiency by reducing redundant computation through the introduction of the Ghost module, and CSPNetv2 [17] reduces computation by partially connecting across stages while retaining more feature information.

To take full advantage of edge computing and cloud computing in defect detection to minimize the resource-constrained problem arising from the limited computing power and storage space of the devices, researchers have proposed a variety of cloud-edge collaborative detection architectures. The authors of [18] propose a sensor network-based cloud-edge cooperative defect detection method, whereby edge computing devices perform initial screening and transmit high-confidence defect data to the cloud for further processing, thus improving detection accuracy and response speed. The authors of [19] propose a cloud-edge cooperative PV module defect detection technique based on cloud-edge collaboration. This technique combines the YOLOv3-tiny algorithm and enhances the small target detection capability by introducing the residual module and shallow feature information fusion. The authors of [20] propose an industrial defect detection system based on cloud-edge collaboration and task scheduling techniques. The system includes a defect detection module, a resource awareness module, a task scheduling module and a model training optimization module. With this architecture, the system can improve the detection quality while reducing the pressure on network and computational resources. The authors of [21] propose a distributed real-time target detection framework based on cloud-edge collaboration for intelligent video surveillance, aiming to reduce bandwidth requirements and improve real-time by performing preliminary data processing through edge computing. The framework employs cloud computing to undertake complex analysis and processing, thereby generating global knowledge that is remotely deployed to edge devices to attain high efficiency and accuracy in intelligent surveillance. Although existing research has made significant progress in circuit board component defect detection and efficient model deployment on resource-constrained devices, there are still some shortcomings. For example, existing small-target detection models still suffer from false and missed detections when dealing with complex backgrounds and high-density targets; lightweight models are deficient in feature extraction capability and detection accuracy in high-resolution images and complex scenes; and cloud-edge collaborative architectures are still faced with challenges such as limited computational power and data transmission delays in real-world deployments.

To address these problems, this paper proposes a lightweight model based on YoloV5n, which usesGhostConv, SCDown, and RepNCSPELAN4 modules for efficient feature extraction and computation, and this paper abbreviates this model as SGRS-YoloV5n. The model maintains high detection accuracy while significantly reducing the model size and computation, making it more suitable for deployment on edge devices. This paper designs a cloud-edge defect detection system to improve accuracy and efficiency through collaboration between edge devices and cloud servers. Specifically, edge devices perform initial defect detection, screen high-confidence samples, and upload low-confidence samples to the cloud for further inspection. The cloud server uses a high-precision model for secondary detection of the uploaded samples and feeds the results back to the edge device, thus realizing efficient and accurate real-time defect detection. Additionally, an incremental learning mechanism is also introduced, enabling the model to adapt promptly to new defect categories and update its parameters accordingly. The main contributions of this paper are as follows:(1)The integration of modules such as GhostConv, SCDown, and RepNCSPELAN4 significantly reduces model size and computational load while achieving efficient feature extraction and computational performance. The SGRS-YoloV5n model demonstrates excellent detection accuracy and real-time performance in actual production environments, proving its application value and stability in resource-constrained settings.(2)The design of a collaborative defect detection system leverages the strengths of both edge devices and cloud servers, improving detection accuracy and efficiency and addressing the limitations of relying solely on edge devices or cloud computing.(3)When the labels of the new training data are in conflict with the knowledge acquired by the pre-trained model, the incremental learning strategy designed directionally updates the inconsistent knowledge of the model with the new training data, thereby achieving beneficial selective forgetting.

## 2. Materials and Methods

### 2.1. Overall Architecture

This paper proposes SGRS-YoloV5n, a lightweight YoloV5 model, to address defect detection in resource-constrained environments. It integrates GhostConv, SCDown, RepNCSPELAN4, and ScalSeq modules to enhance feature extraction and performance while keeping a compact size. The architecture is detailed below.

#### 2.1.1. Cloud-Edge Collaborative Defect Detection System

Edge devices and cloud servers work together for efficient, low-latency, real-time defect detection. Edge devices (Raspberry Pi 4 (Raspberry Pi Foundation, Cambridge, UK) units: A, B, and C) handle initial detection and screen high-confidence samples. Low-confidence samples are sent to the cloud for further inspection. The decision to choose to use three Raspberry Pis was based on optimizing real-time processing and dealing with different detection confidence levels, which is crucial in pipeline scenarios. Due to the limited processing capacity of a single Raspberry Pi and the varying computational complexity of the tasks, modularizing the tasks and assigning them to the three devices, this paper hopes to achieve: (a) load distribution and parallel processing, (b) real-time response to the tasks, and (c) efficient use of resource efficiency, with the tasks of each device configured according to the specific needs of the detection process, ensuring the scalability and efficiency of the system.

Figure 1 shows the cloud-edge collaborative defect detection system.

(1)Raspberry Pi Aperforms initial defect detection with SGRS-YoloV5n. High-confidence samples are processed locally, and a signal is sent to Raspberry Pi C. Low-confidence samples are sent to Raspberry Pi B for re-evaluation, improving detection accuracy and efficiency.(2)Raspberry Pi B uses algorithms for small-target detection, such as multi-scale Feature Extraction, attention mechanisms, etc., to further analyze low-confidence samples from Raspberry Pi A. Uncertain results are sent to the cloud for review. Reviewed components are classified as defective or not.(3)The cloud uses a more accurate model, such as YoloV5l/X, Faster R-CNN, etc., to review results from Raspberry Pi B. If defects are confirmed, the cloud signals Raspberry Pi C for sorting. Non-defective components do not trigger sorting.

#### 2.1.2. Lightweight YoloV5 Model Design

The SGRS-YoloV5n model design includes several key modules:(1)GhostConv and C3 Ghost improve feature extraction by reducing redundant computations, keeping the model lightweight. The SCDown module preserves detailed features and reduces channels by convolution, enhancing feature extraction and computational efficiency.(2)The RepNCSPELAN4 module enhances feature extraction and network performance through convolutional and residual operations, combining CSPNet and ELAN for better feature fusion and multi-scale extraction.(3)The ScalSeq module ensures complete and consistent feature representations through multi-scale fusion. The attention-model module enhances important features and suppresses irrelevant ones, improving detection accuracy and efficiency.

#### 2.1.3. Incremental Learning Mechanism

To ensure the model can adapt to new defect categories efficiently, the incremental learning mechanism is incorporated. This mechanism involves:(1)**Feature Extraction and Similarity Calculation:** New defects are compared with known categories using cosine similarity. Low-similarity samples are labeled as new categories and saved for training.(2)**Freezing Network Update Model:** When a new defect category is detected, the SGRS-YoloV5n model is fine-tuned by freezing all but the last few layers and training only the unfrozen layers. This method ensures the model retains its recognition ability for existing categories while adapting to new ones.

### 2.2. Defect Detection with SGRS-YoloV5n

The overall structure of SGRS-YoloV5n is shown in Figure 2, including three key modules: backbone module based on SCDown and GhostConv, neck module based on RepNCSPELAN4 and GhostConv, and feature fusion and attention mechanism module based on ScalSeq and attention-model. In order to ensure real-time performance and stability on edge devices, this study introduces the backbone module based on GhostConv and C3Ghost, the neck module based on RepNCSPELAN4 and ScalSeq, and the feature fusion and attention mechanism module based on ScalSeq and attention-model in order to improve the feature extraction efficiency and reduce redundant computation while fusing multi-scale features to enhance small-target detection. In this section, this paper will carefully introduce the role and working principle of each module.

(1)**Backbone Design:** Due to the small size of defects in components, features are easily lost during downsampling; important detailed information may be lost during multilayer convolution and downsampling, resulting in lower detection accuracy; and efficient computational power and low resource consumption are required to run on edge devices. To address these challenges, this paper introduces the GhostConv and C3Ghost modules in the backbone part of the network architecture of YoloV5 and adopts the SCDown module. GhostConv and C3Ghost are shown in Figure 3 and Figure 4.

The GhostConv module is a lightweight convolutional operation proposed in GhostNet [16], which aims to reduce the computational and parametric counts while maintaining the expressive power of the model. The core idea is to reduce the computational complexity of traditional convolutional operations by generating more ghost feature maps. Conv is used to generate the initial feature maps:(1)Y=Conv(X,W),
where W is the convolution kernel parameter for regular convolution, Y is the feature generated by regular convolution, and the CheapOperation operation operates on a portion of the initial feature map generated to produce a “shadow” feature map:(2)$$\hat{Y}=T\left(Y\right),$$
where T denotes the cheap operation (e.g., point-by-point convolution, element-by-element weighting, etc.), which is used to generate the “shadow” feature map. Finally, the feature map generated by the conventional convolution Conv and the “shadow” feature map generated by CheapOperation are combined together to obtain the final output feature map:(3)$$Z=[Y,\hat{Y}],$$
where [·] denotes the splicing operation of the feature map. The final output feature map Z is the conventional convolutional feature map Y spliced with the “virtual shadow feature map”.

C3Ghost is a lightweight module proposed in the literature [16], which is a variant of the C3 module, which mainly utilizes GhostBottleneck for feature extraction and fusion. The input tensor $X\in R^{C_{1}\times H\times W}$ is passed through a convolutional layer to obtain the intermediate feature map Y. Each GhostBottleneck module expands the feature map to a high dimensional space and then compresses it back to the original dimensions. The output feature map of each GhostBottleneck module is spliced with Y to obtain the feature map $Y_{\mathrm{concat}}$, and then the output feature map $Y_{\mathrm{output}}$ is obtained through a convolutional layer. The above operation can be expressed as:(4)$$Y^{\prime }=\mathrm{Conv}(\mathrm{DWConv}\left(\mathrm{conv}\left(Y\right)\right),$$
(5)$$Y_{\mathrm{output}}=\mathrm{Conv}(Y_{\mathrm{concat})},$$

The structure of SCDown is shown in Figure 5. The SCDown module is a module proposed in YoloV10 [22] for feature extraction, which reduces the number of channels by a 1 × 1 convolutional layer on the input tensor $X\in R^{C_{1}\times H\times W}$. The 3 × 3 deep convolutional layers perform convolutional operations independently within each channel, reducing the computational cost to $O(2{\mathrm{HWC}}^{2}+9/2\mathrm{HWC})$. The number of parameters is reduced to $O(2C^{2}+18C)$. While maximizing the retention of information in the down-sampling process, the overall operation process of the module can be expressed as follows:(6)$$Y_{1}={\mathrm{Conv}}_{1\times 1}\left(X\right),$$
(7)$$Y_{2}={\mathrm{DepthwiseConv}}_{3\times 3}\left(Y_{1}\right),$$
(8)$$Y_{\mathrm{output}}=Y_{2}+X_{\mathrm{downsampled}},$$

By introducing SCDown, C3Ghost, and GhostConv in the backbone, efficient feature extraction and retention can be achieved, significantly reducing computation and parameter count while enhancing feature expression and multi-scale feature fusion capabilities.

(2)**Neck Design:** In order to strike a balance between the accuracy and complexity of defect detection for small targets in components and to solve the problems of the above-mentioned attention mechanisms, this paper improves the neck part and proposes a neck module based on RepNCSPELAN4 [23] and ScaleSeq. This aims to improve the detection accuracy of the model through efficient feature extraction and fusion, enhanced feature representation, and multi-scale feature processing and real-time performance. The structures of the RepNCSPELAN4 and ScalSeq modules are shown in Figure 6 and Figure 7.

In this module, the input feature map $X\in R^{C_{1}\times H\times W}$ is reduced by a 1 × 1 convolutional layer to obtain the feature map Y. Then, the feature map is chunked into two parts, $Y_{1},Y_{2}$, and then the feature map $Y_{3},Y_{4}$ is obtained through two RepNCSP modules and convolutional layers for deep feature extraction. The feature map $Y_{1},Y_{3},Y_{4}$ is spliced in a channel dimension to obtain the integrated feature map $Y_{\mathrm{concat}}$, which is obtained by a 1 × 1 convolutional layer to obtain the final output feature map. The above operation is formulated as follows:(9)$$\begin{array}{cc} Y & ={\mathrm{Conv}}_{1\times 1}\left(X\right),\\ Y_{1},Y_{2} & =\mathrm{Chunk}(Y,\mathrm{chunks}=2,\dim=1),\\ Y_{3} & ={\mathrm{Conv}}_{3\times 3}\left(\mathrm{RepNCSP}\left(Y_{2}\right)\right),\\ Y_{4} & ={\mathrm{Conv}}_{3\times 3}\left(\mathrm{RepNCSP}\left(Y_{3}\right)\right),\\ Y_{\mathrm{concat}} & =\mathrm{Concat}(Y_{1},Y_{3},Y_{4},\dim=1),\\ Y_{\mathrm{output}} & ={\mathrm{Conv}}_{1\times 1}\left(Y_{\mathrm{concat}}\right). \end{array}$$

Through these operations, the RepNCSPELAN4 module is able to effectively fuse multi-scale features and enhance the comprehensiveness and consistency of feature expression, thus improving the accuracy and robustness of detection.

The ScalSeq module performs multi-scale feature fusion to ensure the comprehensiveness and consistency of the feature representation. The input feature maps are formed into feature maps of the same dimensions after 1 × 1 convolution and up-sampling. These feature maps are expanded into a 3D tensor and spliced in the third dimension, and the fused feature maps are obtained by 3D convolution, batch normalization, and activation processing, followed by 3D pooling and dimensional compression.

### 2.3. Incremental Learning Mechanism

In traditional defect detection systems, models are usually trained with a large volume of labeled data. However, when new classes of defects appear, retraining the whole model is both time-consuming and laborious. To this end, this paper designs an incremental learning method based on feature extraction and cosine similarity computation, which enables the model to adapt and update rapidly for detecting new defect categories.

#### 2.3.1. Feature Extraction and Similarity Calculation

First, this paper extracts high-dimensional feature vectors from the images using the pre-trained EfficientNet-B0 model and saves these feature vectors as “.npy” files for subsequent use. When defective samples with confidence below a threshold are detected, their feature vectors are extracted and compared with a library of known category features, and their similarity to known categories is measured by cosine similarity. The cosine similarity is calculated as:(10)$$C_{A,B}=\frac{A\;\cdot \;B}{∥A∥∥B∥}=\frac{\sum_{i-1}^{n}A_{i}B_{i}}{\sqrt{\sum_{i-1}^{n}A_{i}^{2}}\sqrt{\sum_{i-1}^{n}B_{i}^{2}}},$$

A and B are two feature vectors. When the similarity is below a certain threshold, the sample is considered to belong to a new unknown category and is saved for subsequent training.

In the real-time inspection process, this paper uses the ONNX model for defect detection. The video stream captured by the camera is preprocessed and fed into the ONNX model for inference, and the inference results are filtered by the non-maximal value suppression algorithm to find the optimal detection frame. When a new category sample is detected, its feature vector is extracted and saved, and cosine similarity is computed with a library of known category features to confirm its category or mark it as a new category.

#### 2.3.2. Fine-Tuning Strategy for Knowledge Conflict

Due to the conflict between old and new knowledge, direct fine-tuning with data containing new knowledge frequently proves ineffective. Referring to [24], this paper embraces a paradigm of knowledge renewal, namely, forgetting before learning, and implements it in defect detection. The core concept lies in fine-tuning the initial model with old knowledge, and subsequently subtracting the disparity between the parameters of the fine-tuned model and those of the initial model from the initial model parameters, a procedure designated as “old knowledge forgetting”. Then, the new knowledge is employed to fine-tune the model after the old knowledge is forgotten to achieve the learning of the new defect category. After two phases of forgetting the old knowledge and learning the new knowledge, the knowledge of the model is updated. During the stage of forgetting old knowledge, for a given defect detection model $f_{\theta }$ and its parameters θ, the incremental parameters as knowledge parameters $\theta_{\Delta }$ can be defined as follows:(11)$$\theta_{\Delta }=\mathrm{FT}\left\{\theta ,K\right\}-\theta$$
where FT is the operation of supervised fine-tuning, while K, θ refers to the dataset of knowledge and the parameters of the original model $f_{\theta }$, respectively. Furthermore, a new model $f_{\theta^{\prime }}$ with its parameters $\theta^{\prime }$ can be calculated as follows:(12)$$\theta^{\prime }=\theta -\lambda \theta_{\Delta }$$
where λ is a hyperparameter to control the rate of forgetting. This paper can employ LoRA (Low-Rank Adaptation) to fine-tune the model $f_{\theta }$ and acquire the model $f_{\theta^{\prime }}$ that has forgotten the old knowledge in contrast to $f_{\theta }$. During the stage of learning new knowledge, as the conflict between old and new knowledge has been resolved, the detection of new defect categories can be achieved through the general fine-tuning approach.

## 3. Experimental Analysis

### 3.1. Experimental Environment

To verify our proposed method’s feasibility, this paper used the platform established in the paper [25], as shown in Figure 8. Meanwhile, this paper used the Ubuntu system, PyTorch version 1.10.1 as the deep learning framework, and Yolov5n as the basic network. The experimental environment configuration is shown in Table 1.

The hyperparameter configurations used during training are shown in Table 2.

### 3.2. Datasets and Evaluation Indicators

This paper used a public synthetic PCB dataset from the Open Laboratory for Intelligent Robotics at Peking University. It contains 1386 images with 6 defect types (Missing-hole, Mouse-bite, Open-circuit, Short, Spur, and Spurious-copper), shown in Figure 9.

The dataset was expanded to 6237 images by enhancing brightness, contrast, rotating, and flipping the images. The defect types are in a 1:1:1:1:1:1 ratio, with the training, test, and validation sets in an 8:1:1 ratio.

To evaluate the SGRS-YoloV5n network’s defect detection performance, this paper used Precision, Recall, and mean Average Precision (mAP) as metrics, CIOU for bounding box regression loss, BCELoss for confidence loss, and Cross-Entropy Loss for classification loss. The evaluation formulas are:(13)$$\begin{array}{cc} \mathrm{Precision} & =\frac{\mathrm{TP}}{\mathrm{TP}+\mathrm{FP}}\\ \mathrm{Recall} & =\frac{\mathrm{TP}}{\mathrm{TP}+\mathrm{FN}}\\ \mathrm{mAP} & =\frac{1}{N}\sum {\mathrm{AP}}_{i} \end{array}$$
where N denotes the category overview, TP denotes the number of correctly identified positive samples, FP denotes the number of false positive negative samples, FN denotes the number of missed positive samples, and TN denotes the number of correctly identified negative samples. AP and mAP denote the single class accuracy and average accuracy, respectively. The core formula of the loss function is as follows:(14)$$\begin{array}{c} {\mathrm{bbox}}_{\mathrm{Loss}}=1-\mathrm{IOU}\left(b,b^{*}\right)+\frac{\rho^{2}\left(b_{c},b_{c}^{*}\right)}{c^{2}}+\alpha \;\cdot \;v\\ {\mathrm{obj}}_{\mathrm{Loss}}=-\left(p^{*}\log\left(p\right)+\left(1-p^{*}\right)\log\left(1-p\right)\right)\\ {\mathrm{cls}}_{\mathrm{Loss}}=-\sum_{c=1}^{C}p_{c}^{*}\log\left(p_{c}\right)\\ L_{\mathrm{all}}=\lambda_{\mathrm{bbox}}{\mathrm{bbox}}_{\mathrm{Loss}}+\lambda_{\mathrm{obj}}{\mathrm{obj}}_{\mathrm{Loss}}+\lambda_{\mathrm{cls}}{\mathrm{cls}}_{\mathrm{Loss}} \end{array}$$
where *b* and $b^{*}$ denote the predicted bounding box and ground truth bounding box, respectively. IOU$\left(b,b^{*}\right)$ denotes the Intersection over Union between the predicted and ground truth bounding boxes. $\rho^{2}\left(b_{c},b_{c}^{*}\right)$ denotes the squared distance between the centers of the predicted and ground truth bounding boxes. *c* represents the diagonal length of the smallest enclosing box that covers both the predicted and ground truth bounding boxes. α·v represents a shape constraint term, where α is a weight factor and *v* is the difference in aspect ratios between the predicted and ground truth boxes. *p* and $p^{*}$ represent the predicted probability and the ground truth label, where $p^{*}\in \left\{0,1\right\}$. $p_{c}$ denotes the predicted probability for class *c*. $L_{\mathrm{all}}$ is the total loss, composed of bounding box loss, objectness loss, and classification loss. $\lambda_{\mathrm{box}},\lambda_{\mathrm{obj}},\lambda_{\mathrm{cls}}$ are the weighting factors for the bounding box loss, objectness loss, and classification loss, respectively.

### 3.3. Experimental Results

In this study, we conducted detailed ablation experiments to verify the effect of each module on the lightweight YoloV5 model’s performance. The results are shown in Table 3.

Comparisons show that the backbone network with SCDown and GhostConv modules achieves good accuracy and high speed while maintaining low GFLOPS, with an mAP of up to 0.923 and GFLOPS of 3.1. This shows that the combination of SCDown and GhostConv modules can effectively reduce the computational complexity of the model while maintaining a better feature extraction capability. Adding the RepNCSPELAN4 module slightly improves the mAP to 0.924 and reduces GFLOPS to 2.4, showing it reduces computation while improving detection performance. Introducing ScalSeq and attention-model modules significantly increases mAP to 0.938 and recall to 0.902, with higher GFLOPS. These modules improve detection precision and recall. Combining all optimization modules achieves the best performance with 2.7 GFLOPS, mAP of 0.935, precision of 0.971, and recall of 0.900, significantly improving accuracy and robustness while maintaining efficient speed.

As shown in Figure 10, SGRS-YoloV5n’s mAP improves more slowly than YoloV5n in the early stages but catches up later, reaching a similar mAP. It shows a stable enhancement trend without oscillations, achieving good performance and robustness over a long training period.

The confusion matrix evaluates classification performance, showing accuracy, false alarm rate, and omission rate by comparing actual and predicted categories. Diagonal elements indicate correct classifications: the number of samples where predictions match actual categories. Non-diagonal elements indicate misclassifications: the number of samples where predictions do not match actual categories.

The confusion matrices for Yolov5n and SGRS-YoloV5n are in Figure 11. SGRS-YoloV5n has a slightly worse accuracy and miss rate in some categories but maintains lightness. Both networks have the same correct classification rate for Missing-hole and Short defects. For Spur, Mouse-bite, Open-circuit, and Spurious-copper, SGRS-YoloV5n’s rate is 4–7% lower, but overall performance is acceptable for resource-constrained real-time detection.

### 3.4. Model Performance Comparison

To evaluate performance enhancements, this paper compared augmented models with widely used detection models, including Faster R-CNN (two-stage), YoloV5n and its variants, and transformer-based YoloV5n. All of the above models are based on the Yolov5n model, replacing modules in the backbone network. All experiments used the same dataset and conditions. As shown in Table 4, SGRS-YoloV5n is 2.2 MB compared to Faster R-CNN’s 76.1 MB. Its GFLOPs is 2.7, much lower than Faster R-CNN’s 16.87, suggesting high accuracy with reduced computational cost and resource consumption. Compared to other YoloV5 variants, SGRS-YoloV5n greatly reduces model size and computation, outperforming other models in the Missing-hole and Short categories.

These results show that SGRS-YoloV5n excels in accuracy and efficiency, performing well in resource-constrained environments with outstanding stability. It demonstrates excellent accuracy and efficiency in detecting small-target defects.

Finally, SGRS-YoloV5n detects the PCB defect dataset, with the results shown in Figure 12. Different colored boxes represent different defect types.

### 3.5. Cloud-Edge Collaborative Detection

Based on the cloud-edge collaborative detection framework mentioned in Section 2, this paper designs and implements a real-time defect detection system based on the cloud-edge collaborative architecture to validate the performance of the SGRS-YoloV5n model in real applications. The system aims to fully utilize the advantages of edge computing devices and cloud computing resources to achieve efficient and low-latency real-time defect detection.

(1)Raspberry Pi4 is used as Raspberry Pi A and SGRS-YoloV5n network is deployed for initial defect detection; SGRS-YoloV5n is able to identify most of the defective samples efficiently, and a signal will be sent to the Raspberry Pi C when the threshold of the detected defects is greater than 0.6. The results of the detection as well as the sending of the signals are shown in Figure 13.(2)When Raspberry Pi A detects a defect below confidence threshold (based on the impact of different confidence thresholds on system performance, after empirical background and cross-validation, this confidence threshold was set to 0.6), it signals Raspberry Pi B. Raspberry Pi B re-detects the PCB, sending the original image to the cloud server for YoloV5s detection. If a defect is found, the cloud sends a signal to Raspberry Pi C for sorting. Original and cloud detection results are shown in Figure 14.(3)After the server detects the defect, it will send a signal “1” to the Raspberry Pi C, which is connected to the robotic arm, and the Raspberry Pi C will control the robotic arm to move after receiving the signal “1”.

### 3.6. Incremental Learning Experiment Results

In this experiment, this paper used an industrial defect detection dataset containing six known categories and one new category. The known categories are Missing-hole, Mouse-bite, Open-circuit, Short, Spur, and Spurious-copper. New categories are unknown categories discovered and labeled through the real-time inspection process. The main steps of the experiment are as follows:(1)The pre-trained EfficientNet-B0 model extracts features from the image and saves them as .npy files. Figure 15 shows the six extracted defects and the feature distribution after adding the new features.(2)Real-time detection is performed using the ONNX model. Each frame is preprocessed and input into the ONNX model, with results filtered by non-maximum suppression to find the optimal detection frame. Samples below the confidence threshold are compared with known category features using cosine similarity. If similarity is below the threshold, the sample is labeled as a new category and saved.(3)For new categories, the YoloV5 model is fine-tuned by freezing all but the last few layers and training the unfrozen layers. A VariFocal Loss function and Adam optimizer are used to maintain recognition of old data while fine-tuning new data.

In our experiment, this paper first trained the YoloV5 model using a PCB dataset with six defects (Missing-hole, Mouse-bite, Open-circuit, Short, Spur, Spurious-copper). This paper added a seventh defect, Pad-damage, using incremental learning. After evaluating the new validation set, the results in Table 5 show that mAP@0.5 can reach 0.338 after simple training. This demonstrates that incremental learning can adapt to new classes without full retraining, validating its effectiveness in PCB defect detection. Further optimization is needed to balance detection capabilities of new and old categories.

The original network shows the benchmark performance of detection for each defect type without incremental learning.

The incremental learning network (no full fine-tuning) shows the benchmark performance of detection for each defect type when incremental learning is performed without adjusting the number of datasets.

The incremental learning network (full-scale fine-tuning) shows when incremental learning is performed by adjusting the amount of data for each type of defect to be consistent, the detection benchmark performance for each defect type.

The incremental learning network (our method) shows that when incremental learning is performed with a targeted increase in the amount of defect data, it is not easy to train, and with a decrease or no change in the amount of defect data, it is easy to train—the detection benchmark performance for each defect type.

## 4. Conclusions

This paper proposed a real-time defect detection system for circuit board components using a cloud-edge collaborative architecture. This paper designed and optimized a lightweight YoloV5 model with GhostConv, C3Ghost, and SCDown modules, improving feature extraction and fusion while reducing computational and parameter requirements for efficient edge device operation. Our cloud-edge framework improves real-time response by performing initial detection on edge devices and ensures high accuracy and reliability through secondary detection on the cloud server. Optimizing data transmission and processing reduces delay, enhancing system performance and stability. Moreover, the introduction of an incremental learning mechanism allows the model to efficiently adapt to new defect categories without requiring complete retraining. This mechanism ensures that the system remains up-to-date with evolving production environments, where new defect types may emerge. By incorporating incremental learning, the proposed system not only maintains high detection accuracy for known defect types, but also swiftly integrates new categories, enhancing the robustness and adaptability of the defect detection process. Experimental results show that our method has significant advantages in defect detection for circuit board components, achieving high accuracy and fast detection speed, proving its feasibility and effectiveness in practical applications. The deployment and application of the proposed method on resource-constrained devices demonstrate strong competitiveness and adaptability. Future research can further optimize the model structure and collaborative framework to address more complex industrial inspection tasks, as well as explore more sophisticated incremental learning techniques to further enhance the system’s adaptability to dynamic environments. 

## Figures and Tables

**Figure 1 sensors-24-05921-f001:**
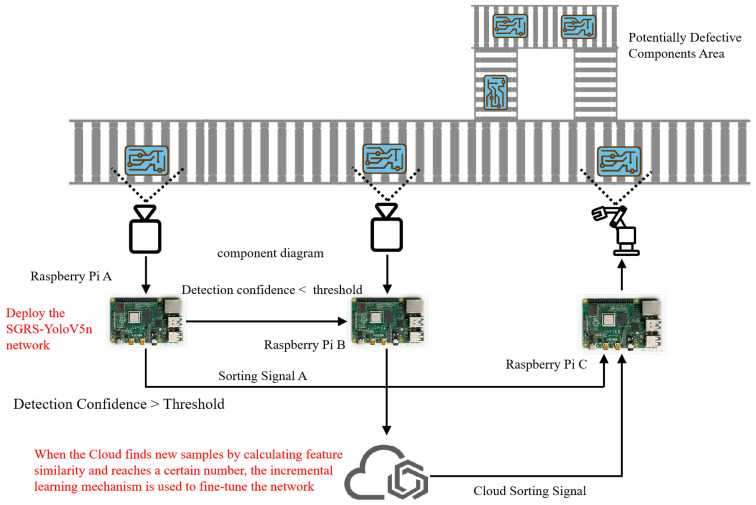
Cloud-edge collaborative defect inspection system.

**Figure 2 sensors-24-05921-f002:**
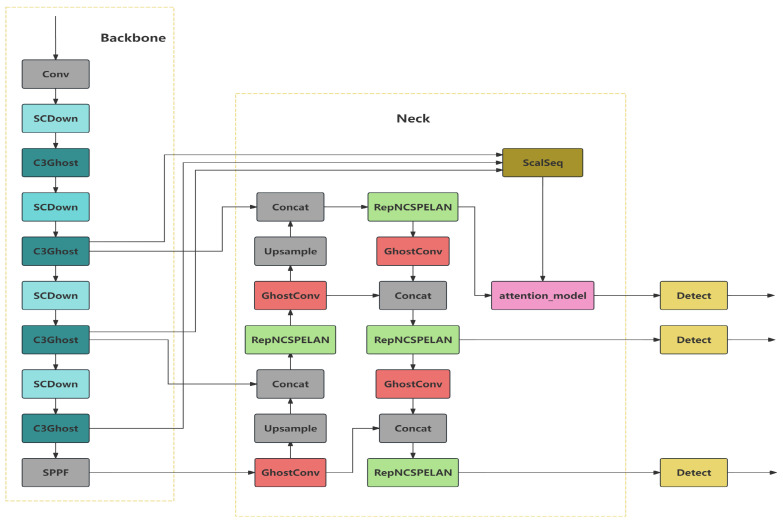
Structure of SGRS-YoloV5n (The model proposed in the article is abbreviated as SGRS-YoloV5n because it is based on the YoloV5n model and combines SCDown, GhostConv, RepNCSPELAN4, and ScalSeq in the backbone and neck parts.).

**Figure 3 sensors-24-05921-f003:**
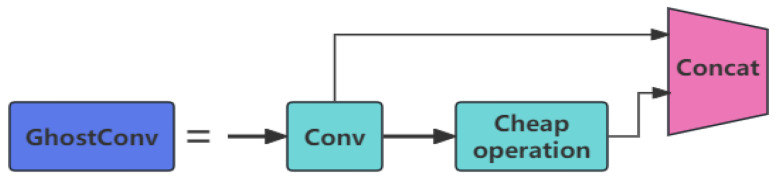
Structure of GhostConv.

**Figure 4 sensors-24-05921-f004:**
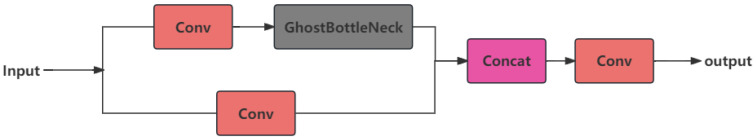
Structure of C3Ghost.

**Figure 5 sensors-24-05921-f005:**
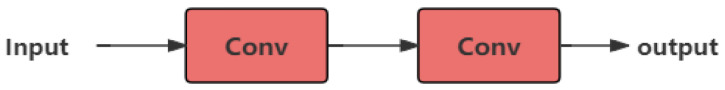
Structure of SCDown.

**Figure 6 sensors-24-05921-f006:**
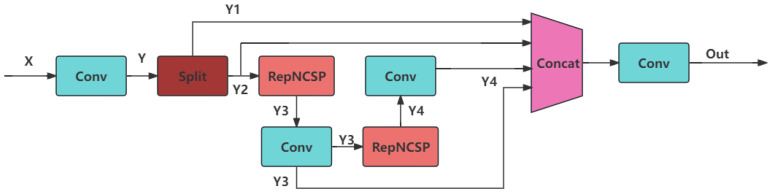
Structure of RepNCSPELAN4.

**Figure 7 sensors-24-05921-f007:**
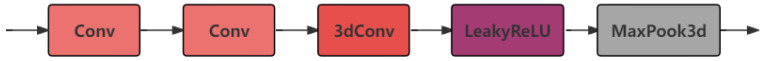
Structure of ScalSeq.

**Figure 8 sensors-24-05921-f008:**
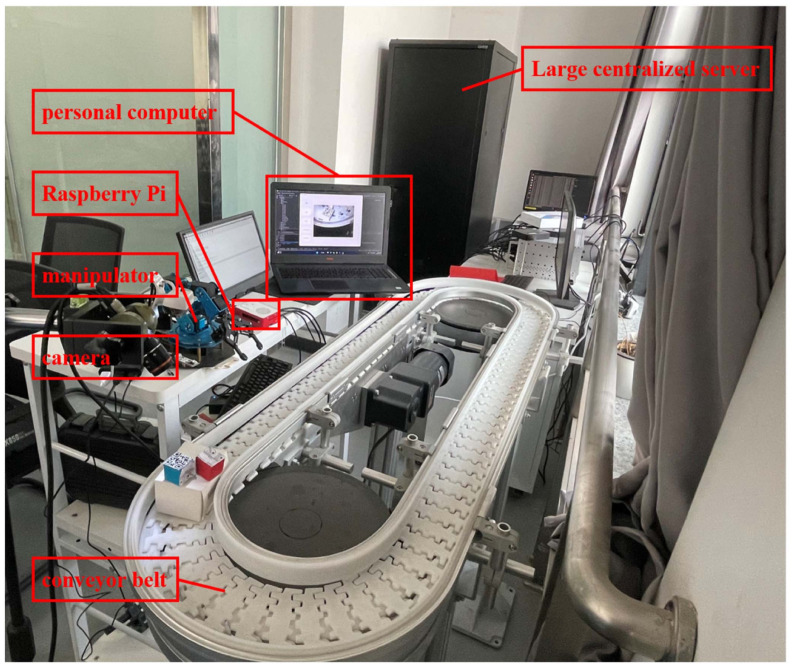
Cloud-edge collaborative defect detection platform.

**Figure 9 sensors-24-05921-f009:**
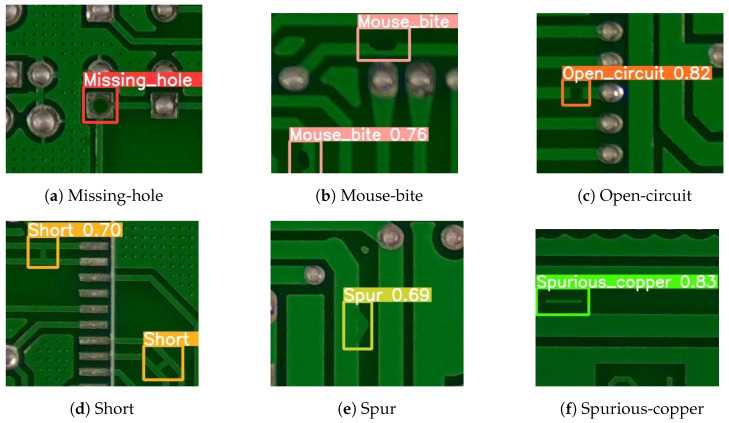
Examples of PCB defect types: (**a**) A defect where a necessary hole is missing. (**b**) Small indentations or nibbles on the PCB edge. (**c**) A break in the circuit where continuity is lost. (**d**) A defect caused by unintended connections between conductive parts. (**e**) An extraneous copper connection leading to an undesired short. (**f**) Unwanted copper residues left on the PCB.

**Figure 10 sensors-24-05921-f010:**
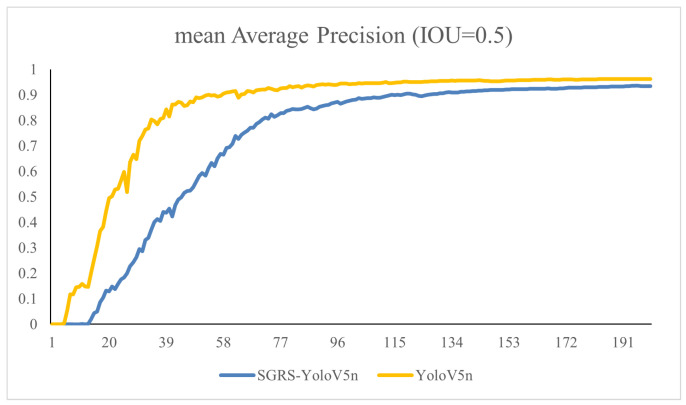
SGRS-YoloV5n and YoloV5n training comparison.

**Figure 11 sensors-24-05921-f011:**
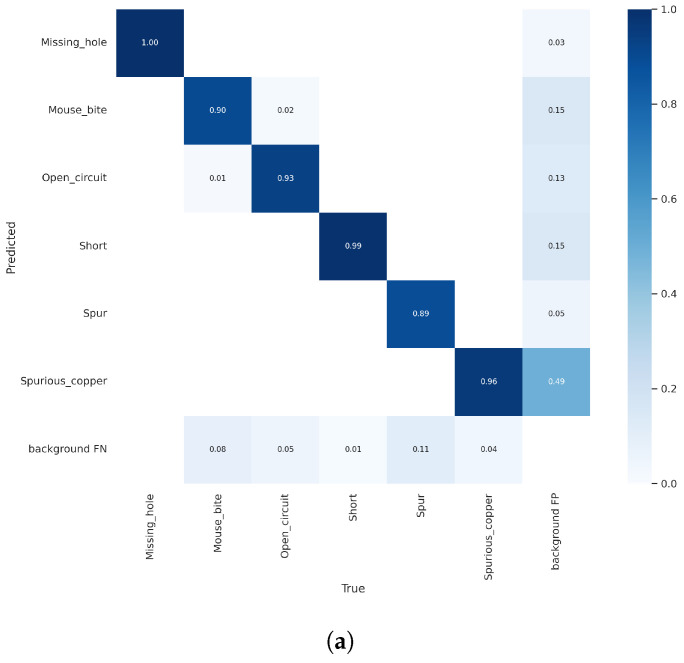
Confusion matrixb comparison. (**a**) Training confusion matrix of Yolov5n; (**b**) training confusion matrix of SGRS-Yolov5n.

**Figure 12 sensors-24-05921-f012:**
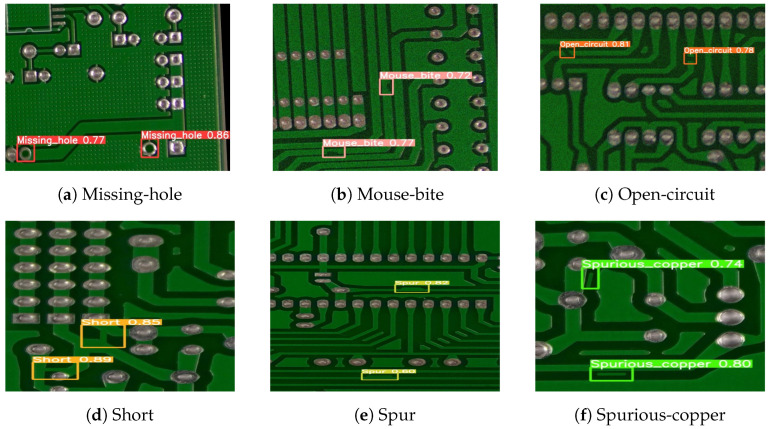
Detection of defect results. (**a**) Detected missing holes with confidence scores of 0.77 and 0.86. (**b**) Detected mouse bites with confidence scores of 0.77 and 0.72. (**c**) Detected open circuits with confidence scores of 0.81 and 0.78. (**d**) Detected shorts with confidence scores of 0.85 and 0.89. (**e**) Detected spurs with confidence scores of 0.82 and 0.60. (**f**) Detected spurious copper with confidence scores of 0.74 and 0.80.

**Figure 13 sensors-24-05921-f013:**
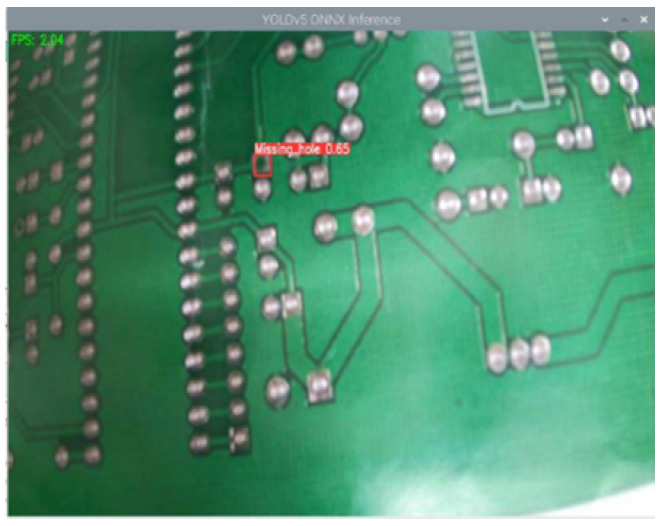
Real-time detection results for edge devices.

**Figure 14 sensors-24-05921-f014:**
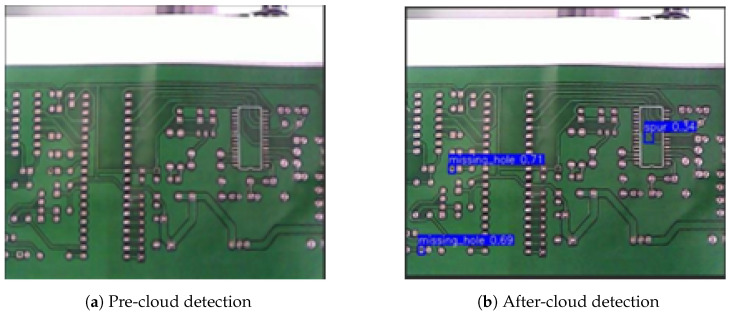
Before and after cloud detecting. (**a**) Original image taken by edge device before cloud processing. (**b**) Detection results after cloud-based processing. The system accurately detects missing holes with confidence scores of 0.71 and 0.69, and a spur with a confidence score of 0.84.

**Figure 15 sensors-24-05921-f015:**
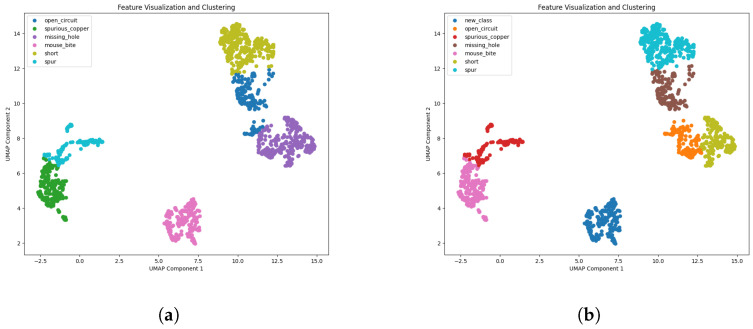
Before and after cloud detecting. (**a**) Original defect characteristics; (**b**) new defective features.

**Table 1 sensors-24-05921-t001:** Experimental environment configuration.

Environment Configuration	Configuration Parameters
System environment	Ubuntu 20.04
Deep Learning Framework	PyTorch 1.10.1
Cuda version	Cuda 11.3
GPU	RTX3090 (24 GB)
CPU	Intel(R) Xeon(R) Silver 4210R CPU @ 2.40 GHz

**Table 2 sensors-24-05921-t002:** Hyperparametric configuration.

Hyperparameter	Value
learning rate	0.01
image size	640 × 640
Momentum	0.937
Batch-size	16
Epochs	200

**Table 3 sensors-24-05921-t003:** Ablation experiment.

Backbone-SCDown-GhostConv	Neck-Rep	Neck-ScalSeq-Attention-Model	P	R	mAP	GFLOPS
✓	-	✓	0.956	0.881	0.923	3.1
✓	✓	-	0.962	0.885	0.924	2.4
-	✓	✓	0.968	0.902	0.938	3.6
✓	✓	✓	0.971	0.900	0.935	2.7

The “-” in the first line indicates whether the proposed module is added to the corresponding structure of the model.

**Table 4 sensors-24-05921-t004:** Comparison experiment.

Model	Defect Types						Size ^1^	GFLOPs
	Missing-Hole	Mouse-Bite	Open-Circuit	Short	Spur	Spurious-Copper		
YoloV5n	0.994	0.879	0.885	0.966	0.844	0.907	8.1	8.9
yolov5n-ShuffleNetv2	0.989	0.782	0.806	0.924	0.779	0.762	29.7	8.0
YoloV5n-MobileNetv3	0.994	0.885	0.928	0.982	0.869	0.923	36.1	7.1
YoloV5n-RepViT	0.993	0.891	0.933	0.976	0.857	0.925	24.6	8.2
YoloV5n-SwinTransformer	0.993	0.883	0.896	0.986	0.835	0.889	222	398.8
YoloV5n-GhostNet	0.989	0.878	0.909	0.979	0.866	0.885	3.0	2.9
YoloV5n-FasterNet	0.994	0.95	0.965	0.988	0.914	0.975	28.8	33.4
Faster-RCNN	0.774	0.894	0.845	0.966	0.815	0.895	74.1	76.1
**SGRS-YoloV5n**	**0.995**	**0.896**	**0.937**	**0.987**	**0.871**	**0.922**	**2.2**	**2.7**

For defect detection metrics (e.g., Missing-hole, Mouse-bite), higher values indicate better performance. ^1^ A smaller size indicates that the model requires less storage space, and smaller GFLOPs indicate that fewer computational resources are consumed.

**Table 5 sensors-24-05921-t005:** Incremental learning experiment.

Network	Missing-Hole	Mouse-Bite	Open-Circuit	Short	Spur	Spurious-Copper	Pad-Damage
original network	0.995	0.896	0.937	0.987	0.871	0.922	-
incremental learning network (No full fine-tuning)	0.722	0.272	0.738	0.938	0.0115	0.580	0.345
incremental learning network (full-scale fine-tuning)	0.808	0.806	0.552	0.923	0.521	0.697	0.338
**incremental learning network** **(our method)**	**0.648**	**0.801**	**0.974**	**0.953**	**0.679**	**0.656**	**0.680**

## Data Availability

The data presented in this study are available on request from the corresponding author.

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
