# Peer review of "Cloud-Edge Collaborative Defect Detection Based on Efficient Yolo Networks and Incremental Learning"

_sensors, 2024, doi:10.3390/s24185921_

Round 1

Reviewer 1 Report

Comments and Suggestions for Authors

The paper proposes a cloud-edge collaboration defect detection based on Raspberry Pi and YOLOv5. The paper is interesting and shows a useful implementation for the field. I have some comments for the authors:

1) My first question is why do you need to use 3 RP? This should be justified in the paper. This analysis should be added to the paper if the justification concerns time processing/consumption.

2) I found a lot of “we” in the paper and generally we avoid writing in the “first person”.

3) The threshold defined was 0.6. Is there any purpose for this value? How did you find this value?

Comments on the Quality of English Language

The quality of English is okay in my opinion. The only issue I found was related to using a lot of first person writing style.

Author Response

Comment1:My first question is why do you need to use 3 RP? This should be justified in the paper. This analysis should be added to the paper if the justification concerns time processing/consumption.

Response1:Thank you for pointing this out. We agree with this comment. In response to this comment, this paper uses edge devices and cloud servers to work together for efficient, low-latency real-time defect detection, a task that can be modularized into three parts: (a) processing local high-confidence detections and sending low-confidence samples; (b) reevaluating the low-confidence samples; and (c) sorting the signals from the cloud and the edge devices; the three parts of the above work to avoid overloading a single device, yet The above three parts avoid overloading a single device and fully utilize the computing resources of each device, ensuring the efficiency and scalability of the system.

In 2.1.1 on page 3 of the manuscript, Edge devices and cloud servers work together for efficient, low-latency real-time defect detection. Edge devices (Raspberry Pi 4 units: A, B, and C) handle initial detection and screen high-confidence samples. Low-confidence samples are sent to the cloud for further inspection. Figure 1 shows the Cloud-Edge Collaborative Defect Detection System. has been changed to Edge devices and cloud servers work together for efficient, low-latency real-time defect detection. Edge devices (Raspberry Pi 4 units: A, B, and C) handle initial detection and screen high-confidence samples. Low-confidence samples are sent to the cloud for further inspection.The decision to choose to use three Raspberry Pi's was based on optimizing real-time processing and dealing with different detection confidence levels, which is crucial in pipeline scenarios. Due to the limited processing capacity of a single Raspberry Pi and the varying computational complexity of the tasks, modularizing the tasks and assigning them to the three devices, we hope to achieve (a) load distribution and parallel processing (b) real-time response to the tasks (c) efficient use of resource efficiency, with the tasks of each device configured according to the specific needs of the detection process, ensuring the scalability and efficiency of the system..

Comment2: I found a lot of “we” in the paper and generally we avoid writing in the “first person”

Response2: Thank you for pointing this out. We agree with this comment. In response to this comment. In the manuscript, all first-person references to we  have been changed to This paper .

Comment3: The threshold defined was 0.6. Is there any purpose for this value? How did you find this value?

Response: Thank you for pointing this out. We agree with this comment.

In our system, the choice of 0.6 as the confidence threshold was arrived at after empirical background and experimental testing. The aim is to ensure sufficiently high detection accuracy without sending too many samples to the cloud for further processing, in order to reduce system latency and optimize resource utilization. If the threshold is set too high , it will result in more samples being considered as low confidence, which will increase the amount of samples sent to the cloud for review, and thus increase the processing latency of the system. Setting too low a threshold can result in too many high-confidence samples being processed incorrectly, reducing the overall detection accuracy of the system. 0.6 as the threshold achieves the optimal detection balance of maintaining high accuracy while minimizing the number of samples requiring further analysis in the cloud.

In (2) of 3.5 on page 15 of the manuscript, “When Raspberry Pi A detects a defect below 0.6, it signals Raspberry Pi B.” has been changed to “When Raspberry Pi A detects a defect below confidence threshold (based on the impact of different confidence thresholds on system performance, after empirical background and cross-validation, this confidence threshold was set to 0.6), it signals Raspberry Pi B.”.

Reviewer 2 Report

Comments and Suggestions for Authors

The topics discussed in the paper are interesting and the results are well presented and discussed. I believe this work is worthy of publication with only minor modifications:

1. the first occurrence of a model abbreviation throughout the article should be written in full, e.g., SGRS-YoloV5n, GhostConv, etc. in the abstract;

2. a space should be added after the punctuation mark in lines 18, 180, 239, and 342, e.g., line 18 should read ".... . Incremental learning; Defect detection; ...";

3. All images in the text should be replaced with clearer images, e.g. in .eps format;

4. Captions of all images in the text should be summarized appropriately;

5. Initial letters of “feature map” and “show that” should be capitalized in lines 227 and 392;

6. Equations (9) to (16) should be integrated and simplified. Simplification of equations (9) to (16) should be integrated;

7. Line 252 .npy should be noted in quotes;

8. Captions for Figures 9, 12 and 14 should include a description of each sub-figure;

9. After equation (23), a necessary description of the symbols in the equation should be added;

10. Tables 3, 4 and 5 should be standardized in the number of significant digits of decimals, e.g., 0.9 should be changed to 0.900;

11. In the section on the comparative experiments of 3.4 of the text, a brief summary of details of the other comparative models used is missing, and is it possible that a brief description of the other comparative models used is missing? In the section on comparative experiments in 3.4 of the text, there is a lack of a brief summary of the details of the other comparative models used, were some of the references used as a basis for the development of these models? If so, please label and cite them in the text and in Table 4;

12. footnotes to the bolded text should be added to Table 4, and to other tables as necessary;

13. footnotes to each metric, e.g., whether certain metrics are better or worse, should be added to Table 4. 14. the progressive learning experiment in Table 5 lacks an explanation of the changes in the metrics in the table.

Comments on the Quality of English Language

The article is readable, but some of the English and formatting could be improved. For more information, see the section on "Comments and Suggestions for Authors
".

Author Response

Comment1: the first occurrence of a model abbreviation throughout the article should be written in full, e.g., SGRS-YoloV5n, GhostConv, etc. in the abstract;

Response1: Thank you for pointing this out. We agree with this comment. In response to This paper addresses the problem of insufficient detection accuracy of existing lightweight models on resource-constrained edge devices by presenting a new lightweight YOLOv5 model, named SGRS-YoloV5n, based on GhostConv, SCDown, RepNCSPELAN, and ScalSeq modules. in the abstract, revise it to This paper addresses the problem of insufficient detection accuracy of existing lightweight models on resource-constrained edge devices by presenting a new lightweight YoloV5 model, which integrates four modules, SCDown, GhostConv, RepNCSPELAN4, and ScalSeq. Here this paper  abbrebriate it as SGRS-YoloV5  in the response manuscript.

On page 3 of the new manuscript, revise To address these issues, this paper propose a new lightweight YOLOv5 model, SGRS- YoloV5n, using GhostConv, SCDown, and RepNCSPELAN4 for efficient feature extraction and computation.  to  To address these problems, this paper proposes a lightweight model based on YoloV5n, which uses GhostConv, SCDown, and RepNCSPELAN4 modules for efficient feature extraction and computation, and this paper abbreviate this model as SGRS-YoloV5n..

Comment2: a space should be added after the punctuation mark in lines 18, 180, 239, and 342, e.g., line 18 should read ".... . Incremental learning; Defect detection; ...";

Response2: Thank you for pointing this out. We agree with this comment. In response to this comment, a space has been added after the punctuation mark lines 18, 180, 239, and 342,e.g..This change is now in lines 18, 190, 241 and 356 of the new manuscript. Problems with spaces after other punctuation mark in the manuscript were also checked and spaces have been added.

Comment3: All images in the text should be replaced with clearer images, e.g. in .eps format;

Reponse3: Thank you for pointing this out. We agree with this comment.. In response to this comment, the unclear images in Figures 9 and 12 have been replaced with clear images in “eps” format.

Comment4: Captions of all images in the text should be summarized appropriately

Response4: Thank you for pointing this out. We agree with this comment.. The summaries of the pictures in Figures 2, 3, 4, 5, 6 and 7 have been replaced with Structure of xxx, e.g. the explanation of Figure 2 has been changed to Structure of SGRS-Yolov5n.

Comment5: Thank you for pointing this out. We agree with this comment. Initial letters of “feature map” and “show that” should be capitalized in lines 227 and 392;

Response5: 

In response to this comment, changes have been made in the article. An . was incorrectly added before feature map in line 227 , after removing it, feature map remains unchanged. This change is now in line 236 of the new manuscript.

The show that in line 392 is consecutive to the preceding Table 5, so the misplaced . has been removed from line 392.This change is now in line 408 of the new manuscript

Comment6: Equations (9) to (16) should be integrated and simplified. Simplification of equations (9) to (16) should be integrated;

Response6: 

Thank you for pointing this out. We agree with this comment. In response to this comment, for equations (9) through (16) on lines 228 through 234, the revised manuscript combines equations (11) and (12) into one equation, and equations (13) and (14) into one equation,with the overall equation (9), and the subsequent equation numbering in the manuscript changes accordingly.

Comment7: Line 252 .npy should be noted in quotes;

Response7: Thank you for pointing this out. We agree with this comment. In response to this comment, in the revised manuscript, line 252 of .npy has been changed to .npy.

Comment8: Captions for Figures 9, 12 and 14 should include a description of each sub-figure;

Response8: Thank you for pointing this out. We agree with this comment. In response to this comment, in the revised manuscript, a summary of each sub-figure has been added to the caption summary in Figure 9 on page 10, which is (a)A defect where a necessary hole is missing.(b)Small indentations or nibbles on the PCB edge.(c)A break in the circuit where continuity is lost.(d)A defect caused by unintended connections between conductive parts.(e)An extraneous copper connection leading to an undesired short.(f)Unwanted copper residues left on the PCB.

A summary of each sub-figure is added to the caption of Figure 12 on page 14, i.e., (a) Detected missing holes with confidence scores of 0.77 and 0.86. (b)Detected mouse bites with confidence scores of 0.77 and 0.72. (c) Detected open circuits with confidence scores of 0.81 and 0.78. (d) Detected shorts with confidence scores of 0.85 and 0.89. (e) Detected spurs with confidence scores of 0.82 and 0.60. (f) Detected spurious copper with confidence scores of 0.74 and 0.80.

The caption in Figure 14 on page 15 adds a summary of each sub-figure, which is “(a) Original image taken by edge device before cloud processing. (b) Detection results after cloud-based processing. The system accurately detects missing holes with confidence scores of 0.71 and 0.69, and a spur with a confidence score of 0.84.”

Comment9: After equation (23), a necessary description of the symbols in the equation should be added;

Response9: Thank you for pointing this out. We agree with this comment. A description of each symbol has been added below equation (14) on page 11 of the new manuscript, which is “where and denote the predicted bounding box and ground truth bounding box, respectively. denotes the Intersection over Union between the predicted and ground truth bounding boxes. denotes the squared distance between the centers of the predicted and ground truth bounding boxes.  represents the diagonal length of the smallest enclosing box that covers both the predicted and ground truth bounding boxes. represents a shape constraint term, where  is a weight factor and  is the difference in aspect ratios between the predicted and ground truth boxes.  and  represent the predicted probability and the ground truth label, where .denotes the predicted probability for class . is the total loss, composed of bounding box loss,  objectness loss, and classification loss.  are the weighting factors for the bounding box loss,  objectness loss, and classification loss, respectively. ”

Comment10: Tables 3, 4 and 5 should be standardized in the number of significant digits of decimals, e.g., 0.9 should be changed to 0.900;

Response10: Thank you for pointing this out. We agree with this comment. In response to this comment, on page 11 of the manuscript, in the fourth line of the fifth column of  Table 3, replace 0.9 with 0.900, and keep the same valid numbers for the same columns in all other tables.

Comment11: In the section on the comparative experiments of 3.4 of the text, a brief summary of details of the other comparative models used is missing, and is it possible that a brief description of the other comparative models used is missing? In the section on comparative experiments in 3.4 of the text, there is a lack of a brief summary of the details of the other comparative models used, were some of the references used as a basis for the development of these models? If so, please label and cite them in the text and in Table 4;

Response11: Thank you for pointing this out. We agree with this comment. In response to this comment, a brief summary of the other models has been added to 3.4 of 13 pages of the manuscript, which is “All of the above models are based on the Yolov5n model, replacing modules in the backbone network.”. Since the replacement modules in the other models are only mentioned in their proposing literature and are not integrated with YoloV5n, there are no relevant references to be used as a basis for building these models, but the literature proposing these modules has been added in the references.

Comment12: footnotes to the bolded text should be added to Table 4, and to other tables as necessary;

Response12: Thank you for pointing this out. We agree with this comment. In response to this comment. In the new manuscript, the text of the footnotes to Tables 1 through 5 has been changed to boldface type.

Comment13: footnotes to each metric, e.g., whether certain metrics are better or worse, should be added to Table 4.

Response13: Thank you for pointing this out. We agree with this comment. In response to this comment. A footnote has been added to Table 4 on page 13 of the novice manuscript, footnote 1 is For defect detection metrics (e.g., Missing holes, Mouse bite), higher values indicate better performance., footnote 2 is A smaller Size indicates that the model requires less storage space, and smaller GFLOPs indicate that less computational resources are consumed.

Comment14: the progressive learning experiment in Table 5 lacks an explanation of the changes in the metrics in the table.

Response14: Thank you for pointing this out. We agree with this comment. In response to this comment. An explanation of the experiments in Table 5 was added on page 16 of the new manuscript, which is Original network shows the benchmark performance of detection for each defect type without incremental learning.

Incremental learning network(No full fine-tuning) shows the benchmark performance of detection for each defect type when incremental learning is performed without adjusting the number of datasets.

Incremental learning network(full-scale fine-tuning) showns that when incremental learning is performed by adjusting the amount of data for each type of defect to be consistent, the detection benchmark performance for each defect type.

Incremental learning network(our method) shows that when incremental learning is performed with a targeted increase in the amount of defect data that is not easy to train and a decrease or no change in the amount of defect data that is easy to train, the detection benchmark performance for each defect type.

Thank you for your review comments in your busy schedule, we agree with your review comments and have responded to each of your review comments in the above section, however, in response to Comment9, Response9 is missing some of the symbols due to the fact that you can't add the symbols of the formulas in the box, there are complete symbols in Response9 in the attached file, please check it out!